# Temporal relationships among role stress, staff burnout, and residents' behavioral problems: A 2-year longitudinal study in child care homes in Hong Kong

Ted C. T. Fong[1], Rainbow T. H. Ho[1,2]*, Joyce C. Y. Fong[1]

**1** Centre on Behavioral Health, The University of Hong Kong, Hong Kong, Hong Kong, **2** Dept of Social Work & Social Administration, The University of Hong Kong, Hong Kong, Hong Kong

* tinho@hku.hk

**Data Availability Statement:** All relevant data are within the paper and its Supporting Information files.

## Abstract

Young residents in care homes experience psychological distress arising from their complex family backgrounds. Residential care workers face job demands and are prone to burnout due to the role stress of balancing enormous workloads with residents' emotional needs. This 2-year study examined the changes in role stress, burnout, and residents' behavioral problems, and their temporal relationships in a sample of 381 young residents and 76 workers from residential care homes in Hong Kong. The workers completed the Role Questionnaire and Copenhagen Burnout Inventory and evaluated the residents' behavioral problems using the Child Behavior Checklist every 3 months. Latent growth modeling was used to analyze the temporal changes, and multilevel regression analysis was used to evaluate the associations between role stress and residents' behavioral problems. The workers displayed stable trends in role stress and burnout with significant inter-individual variations in temporal changes. The residents' total behavioral problems displayed piecewise decreasing trajectories with significant declines over the first 9 months. Controlling for baseline levels, changes in role stress showed significant and positive associations with changes in work burnout and total behavioral problems over the first 9 months. The findings provide support for the temporal relationships among role stress, work burnout, and residents' behavioral problems in a residential care setting.

## Introduction

In Hong Kong, residential child care services are available to children and youths up to the age of 21 whose families fail to provide adequate care. In 2019, over 2,300 children and youths were placed in residential care homes, with the average length of stay ranging from 2 months to 4 years [1]. A further 540 children were waiting for placements, with an average waiting time of 4.6 months. Such adolescents have an elevated risk of experiencing family crises and psychological distress, which may manifest in behavioral problems [2]. Residential care

**Funding:** The second author (RTHH) was supported by the Keswick Foundation Limited in conducting this research study. The Keswick Foundation (http://www.keswickfoundation.org.hk/en/home) is a charitable organization established in 1979 by Sir John Keswick and his daughter, Maggie Keswick Jencks. The funders had no role in study design, data collection and analysis, decision to publish, or preparation of the manuscript.

**Competing interests:** The authors have declared that no competing interests exist.

workers are responsible for managing the family and welfare issues of the residents and providing them with therapeutic residential care [3]. The hallmark of therapeutic residential care is the constant endeavor to uphold essential and strong family links. Work overload is common given the high demand for residential child care services.

Residential child care is a form of human service work that is labor intensive and involves emotionally charged interactions between care workers and young residents. Child care workers may suffer from emotional exhaustion due to job burnout [4, 5]. Burnout is an adverse work phenomenon caused by occupational factors such as work overload, lack of control, and organizational inequality [6] and is associated with depersonalized work attitudes toward clients [7]. The job demands–resources model [8] states that prolonged experience of job demands and the absence of job resources will result in negative work states and job burnout. According to this model, role ambiguity and role conflict are job demands that could contribute to chronic stressors and burnout in the long run [9]. Role ambiguity refers to the uncertainty around accurately assessing different types of tasks and responsibilities in relation to goals and expectations [10], while role conflict arises when workers are confronted with inconsistent or incompatible demands at work [11].

Previous studies have found role conflict and role ambiguity to be positively associated with burnout among various human service professionals such as US elderly care workers [12], Chilean health care workers [13], Indian nurses [14], and US mental health workers [15]. In the residential care setting, workers are expected to simultaneously perform remedial, preventive, and developmental functional roles for the child residents [16]. A preoccupation with administrative and parent-oriented tasks could lead to role conflict and role ambiguity. A previous study [17] found higher levels of workloads, role conflicts, and depersonalization among child welfare workers than among social workers in other fields.

Despite the consistent relationship between role stress and burnout in the literature, previous studies have generally used cross-sectional observational designs that could not elucidate the temporal relationships between the variables. An exception is the longitudinal study of US child welfare workers conducted by Travis, Lizano [18], which found that role conflict, but not role ambiguity, predicted emotional exhaustion 6 months later. Further longitudinal studies with multiple follow-up points are needed for a more precise understanding of the long-term developmental trajectories of role stress and job burnout and the temporal relationships between their trajectories.

Previous studies [19, 20] have revealed behavioral problems such as aggressivity, delinquency, and attention problems among residents of child care homes. Counseling interventions are needed to tackle residents' emotional issues and to ameliorate their associated behavioral problems. A preoccupation with administrative tasks and inconsistent work demands can hinder workers' effective handling of in-depth emotional issues [4]. The prolonged presence of role stress may reduce workers' motivation and efficacy in daily operations, leading to disruptions in service delivery and potential deteriorations in service quality [21, 22]. Over time, these adverse effects could be manifested in terms of worsening behavioral problems among the young care home residents. To the best of our knowledge, no study has yet examined the temporal associations between workers' role stress and residents' behavioral problems.

In light of the abovementioned research gaps, the present study had several research objectives. The first objective was to assess the developmental trajectories of role stress and burnout among child care workers over a 2-year period and to model the developmental trajectories of behavioral problems among the child residents. It was expected that inter-individual variations would exist in the trajectories of the workers and child residents. The second objective was to examine the temporal relationship between the workers' role stress and burnout. Role

ambiguity and role conflict were expected to display longitudinal associations with job burn-out. The third objective was to evaluate the temporal associations between changes in the workers' role ambiguity and role conflict and changes in the residents' behavioral problems via multilevel regression. The investigation was designed to contribute to a better understanding of how job demands in the residential care home workplace affect the well-being of clients.

## Methods

### Participants and procedures

This study adopted a 2-year longitudinal design with a measurement time lag of 3 months. Data collection took place between April 2016 and March 2019 in four residential care homes run by non-governmental organizations in Hong Kong. All workers in the residential care homes were invited to join the study. The residents were eligible for inclusion if they were aged between 6 and 18 years and able to speak Cantonese. Those who had been diagnosed with a major psychiatric disorder were excluded. A total of 409 youth residents and 113 work-ers were invited to join the study, of whom 381 youth residents (response rate = 93.2%) and 76 workers (response rate = 67.3%) agreed to participate. More than half (56.6%) of the residents were female and the mean sample age was 12.1 years (*SD* = 3.2). Most of the workers were female (88.4%), and their mean age was 46.9 years (*SD* = 10.4 years). Their average length of work experience in the social service field was 9.1 years (*SD* = 7.8 years).

### Ethical considerations

The child care workers provided voluntary written informed consent prior to joining the study. Assent was obtained from the child residents, and written informed consent was pro-vided by their parents or guardians. The consent and assent forms explained the study proce-dures in terms of the duration, nature, and number of assessments. To facilitate the decision-making process, the child assent forms were at an appropriate level of comprehensibility for their age and reading level. No incentives were offered to the participants. Data confidentiality was ensured by storing the collected questionnaires in locked cabinets and encrypting the data file with personal identifiers. The participants could withdraw from the study at any time with-out any consequence. Ethical approval was obtained from the Human Research Ethics Com-mittee of the University of Hong Kong (Reference no.: EA1603044).

### Measures

The Role Questionnaire [23] was used to measure the role ambiguity and role conflict of the residential care home workers at baseline (Time 1). Follow-up assessments were conducted every 3 months thereafter until the 18-month follow-up (Time 7) or the worker left the resi-dential home. Sample items for the role ambiguity subscale (6 items) include "I feel certain about how much authority I have" and "I know what my responsibilities are," and sample items for the role conflict subscale (8 items) include "I work with two or more groups who operate quite differently" and "I work on unnecessary things." Each item is scored on a 7-point Likert scale (1 = "very false" to 7 = "very true"), and all items on the role ambiguity subscale are reverse scored. The aggregate scores on the role ambiguity and role conflict sub-scales were calculated as the average of the item scores. This scale has been validated among high school counselors [24]. In the present study, both role ambiguity and role conflict demon-strated good levels of reliability ($\alpha$s > 0.80) across the seven measurement waves.

The Copenhagen Burnout Inventory [25] was used to measure the workers' burnout every 3 months from baseline up to the 24-month follow-up (Time 9). This inventory evaluates the

degree of physical and emotional exhaustion experienced by workers with regard to their work or clients. Sample items for work burnout (7 items) include "are you exhausted in the morning at the thought of another day at work?" and "does your work frustrate you?" and sample items for client burnout (6 items) include "does it drain your energy to work with clients?" and "are you tired of working with clients?" The items are scored on a 5-point Likert scale (0–25–50–75–100, where 0 = "never/to a very low degree" to 100 = "always/to a very high degree") and averaged to produce global scores for work and client burnout. The scale has demonstrated good reliability and factorial validity in the Chinese context [25] and displayed good levels of reliability ($\alpha$s > 0.80) across the nine waves of measurements in this study. A score of 50 or above has been suggested as a criterion for a high level of burnout [26].

The residents' behavioral problems were assessed using the Chinese version of the Child Behavior Checklist [27]. The assessment was conducted at baseline when the residents entered the care home, followed by subsequent assessments every 3 months until the end of the project at the 27-month follow-up (Time 10) or the residents left the residential units. The assessment was performed by the corresponding care worker who was most familiar with the resident. The scale comprises 112 items covering various problematic behaviors such as withdrawn behavior, anxious/depressed mood, attention problems, delinquent behavior, and aggressive behavior, and is widely used in research and clinical practice with youths. All of the items were scored on a 3-point Likert scale (0 = "never/to a very low degree" to 2 = "substantial/to a very high degree") and summed to produce an aggregate score for total behavioral problems. The scale displayed excellent levels of reliability ($\alpha$s > 0.90) across the 10 waves of assessments.

## Data analysis

To account for dropouts in this longitudinal study, attrition analysis was conducted to compare the baseline profiles between the workers who remained at the end of the study and the dropouts. The baseline profiles of the residents were also compared across the duration of their stay in the residential care home. Missing data were handled via full information maximum likelihood under the missing at random assumption [28], enabling the use of all available data in the model estimation. The dataset analyzed in the present study is available in the S1 File accompanying the paper.

The temporal changes in staff well-being and residents' behavioral problems were analyzed via latent growth modeling using robust maximum likelihood estimation in Mplus 8.1 [29]. Latent growth analysis is a flexible analytical technique that examines growth trajectories and models intra-individual changes and inter-individual variations across time [30]. In the present study, the slope factor denotes the change in the variables across 9 months. The collection of multiple measurements over a long follow-up period allowed a precise estimation of the developmental trajectories. Linear, quadratic, and piecewise growth models were fitted to the repeated measures and compared in terms of model fit to the data. The cut-off criteria for the model fit indices [31] included non-significant chi-square ($\chi^2$) ($p > 0.05$); comparative fit index (CFI) $\geq$ 0.95; root-mean square error of approximation (RMSEA) $\leq$ 0.06; and standardized root-mean-square residual (SRMR) $\leq$ 0.08. Model comparison was based on the Bayesian information criterion (BIC), with lower values denoting greater parsimony [32].

The changes in work and client burnout were regressed on the changes in role ambiguity and role conflict at the workers' level. As the study design meant the resident data were nested within the residential care worker data, multilevel regression analysis was conducted to regress the change in residents' behavioral problems on the change in workers' role stress. A number of potential confounding factors were controlled for in the multilevel regression. At the residents' level, gender, age, and the number of completed assessments were included as control

**Table 1. Total response rate across the measurement time points for child care workers and residents in the study.**

|  | Workers sample size | Dropout | Retention rate | Residents sample size | Dropout | Retention rate |
|---|---|---|---|---|---|---|
| Time 1 | 76 | 0 | 100% | 381 | 0 | 100% |
| Time 2 | 71 | 5 | 93.4% | 343 | 38 | 90% |
| Time 3 | 68 | 3 | 89.5% | 300 | 43 | 78.7% |
| Time 4 | 65 | 3 | 85.5% | 238 | 62 | 62.5% |
| Time 5 | 62 | 3 | 81.6% | 205 | 33 | 53.8% |
| Time 6 | 59 | 3 | 77.6% | 147 | 58 | 38.6% |
| Time 7 | 56 | 3 | 73.7% | 111 | 36 | 29.1% |
| Time 8 | 52 | 4 | 68.4% | 101 | 10 | 26.5% |
| Time 9 | 51 | 1 | 67.1% | 89 | 12 | 23.4% |
| Time 10 |  |  |  | 73 | 16 | 19.2% |

variables, and at the workers' level, their associated non-governmental organization, age, gender, work experience, and number of completed assessments were included.

## Results

### Attrition analysis

Table 1 displays the total response rates across the measurement time points for the child care workers and residents. Twenty-five of the 76 recruited workers had dropped out by the time of the final follow-up, implying a retention rate of 67.1% over the 24-month period. The reasons for dropout included change of job and termination of job contract. No significant differences were found ($p$ = .08–.81) in the demographic characteristics and baseline role stress and burnout between the study completers and dropouts. The child residents on average received 5.2 assessments ($SD$ = 3.4) from their care workers. Around half of the sample (46.2%) resided in the residential care homes for less than 1 year and the other half (53.8%) for at least 1 year. A comparison between the residents with short and long stays stay did not reveal any significant differences in their age and total behavioral problems at baseline ($p$ = .14–.61).

### Growth trajectories of role stress and burnout

Table 2 presents the fit indices of the latent growth models for the study variables. The linear growth model (Fig 1) showed a satisfactory fit for role ambiguity and role conflict over the

**Table 2. Model fit indices of the latent growth models for the study variables.**

| Model | $\chi^2$ | $df$ | $p$ | CFI | RMSEA | SRMR | BIC |
|---|---|---|---|---|---|---|---|
| Role ambiguity—Linear growth | 14.7 | 17 | .62 | 1.00 | .000 | .090 | 1441.4 |
| Role conflict—Linear growth | 6.5 | 17 | .99 | 1.00 | .000 | .038 | 1137.0 |
| Work burnout—Linear growth | 49.8 | 39 | .12 | .96 | .060 | .078 | 4409.2 |
| Client burnout—Linear growth | 50.8 | 39 | .10 | .96 | .063 | .075 | 4376.0 |
| Total behavioral problems |  |  |  |  |  |  |  |
| Linear growth | 111.2 | 50 | < .01** | .93 | .057 | .070 | 6720.7 |
| Quadratic growth | 64.3 | 46 | .04* | .98 | .032 | .053 | 6686.9 |
| Logarithmic growth | 63.1 | 46 | .05 | .98 | .031 | .051 | 6684.5 |

* $p$ < .05

** $p$ < .01. $\chi^2$: robust chi-square; $df$: degree of freedom; CFI: comparative fit index; RMSEA: root mean square error of approximation; SRMR: standardized root mean square residual; BIC: Bayesian information criterion.

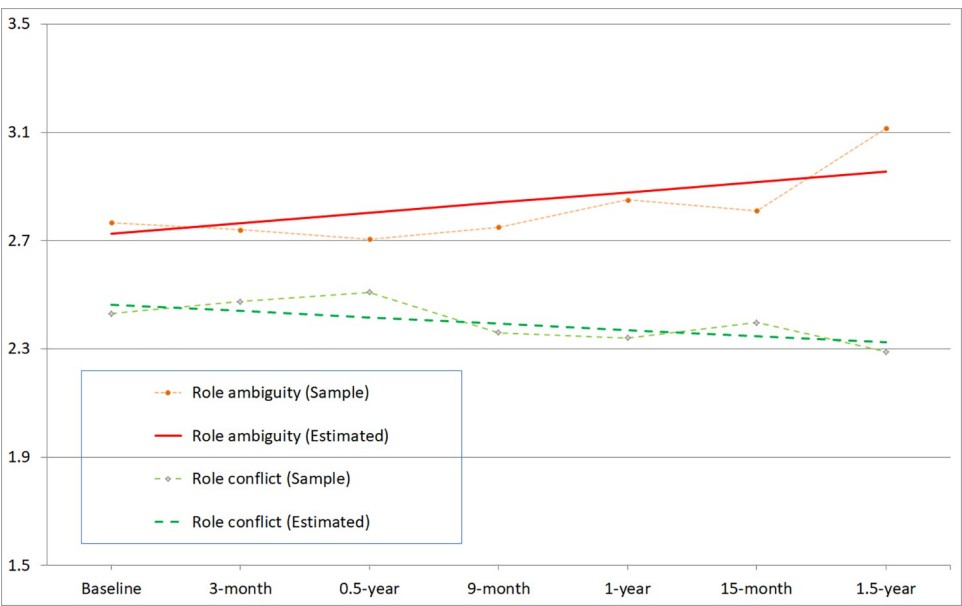

**Fig 1. Developmental trajectories of role ambiguity and role conflict in the child care workers.**

18-month period. The two variables displayed stable trends around their baseline levels (M = 2.7 and 2.5, respectively) with non-significant changes ($\Delta$ = -0.07–0.11, $p$ = .20–.30) over a 9-month interval. The changes in role ambiguity and role conflict showed significant inter-individual variations (SD = 0.33–0.60, $p$ < .01): the variables were moderately correlated at baseline ($r$ = .54, $p$ < .01), but their temporal changes were not significantly correlated ($r$ = .02, $p$ = .95).

The linear growth model (Fig 2) showed a good fit to the work burnout and client burnout data over the 24-month period. Work burnout and client burnout displayed steady trends around their baseline levels (M = 31.6 and 26.4, respectively) with non-significant changes ($\Delta$ = -0.41–0.40, $p$ = .62 –.64) over a 9-month interval. The proportion of child care workers with high levels of burnout remained relatively stable over the 24-month period, ranging from 11.8% to 21.5% for work burnout and from 8.9% to 18.6% for client burnout. The changes in work burnout and client burnout showed significant inter-individual variations (SD = 4.83–4.84, $p$ < .01): the variables were strongly correlated at baseline ($r$ = .87, $p$ < .01), and their temporal changes were strongly associated across time ($r$ = .87, $p$ < .01).

## Developmental trajectories in residents' behavioral problems

The linear growth model showed a mediocre fit to the data on the residents' total behavioral problems. Both quadratic and piecewise growth models provided better model fits with smaller chi-square, higher CFI, and lower RMSEA and SRMR values. The piecewise growth model was chosen as it had the lowest BIC out of the three models. Fig 3 displays the piecewise growth trajectories of total behavioral problems across the 27-month period. From the average starting level of 33.5 (SD = 19.9), total behavioral problems showed a significant decrease over the first 9 months (M = -7.31, $p$ < .01) and a non-significant decrease (M = -1.24, $p$ = .17) over the subsequent 18 months. The temporal changes in total behavioral problems displayed significant inter-individual variations over the first 9 months (SD = 16.2, $p$ < .01) and the following 18 months (SD = 8.1, $p$ < .01). The changes in total behavioral problems across the two phases were not significantly correlated ($r$ = -.20, $p$ = .21).

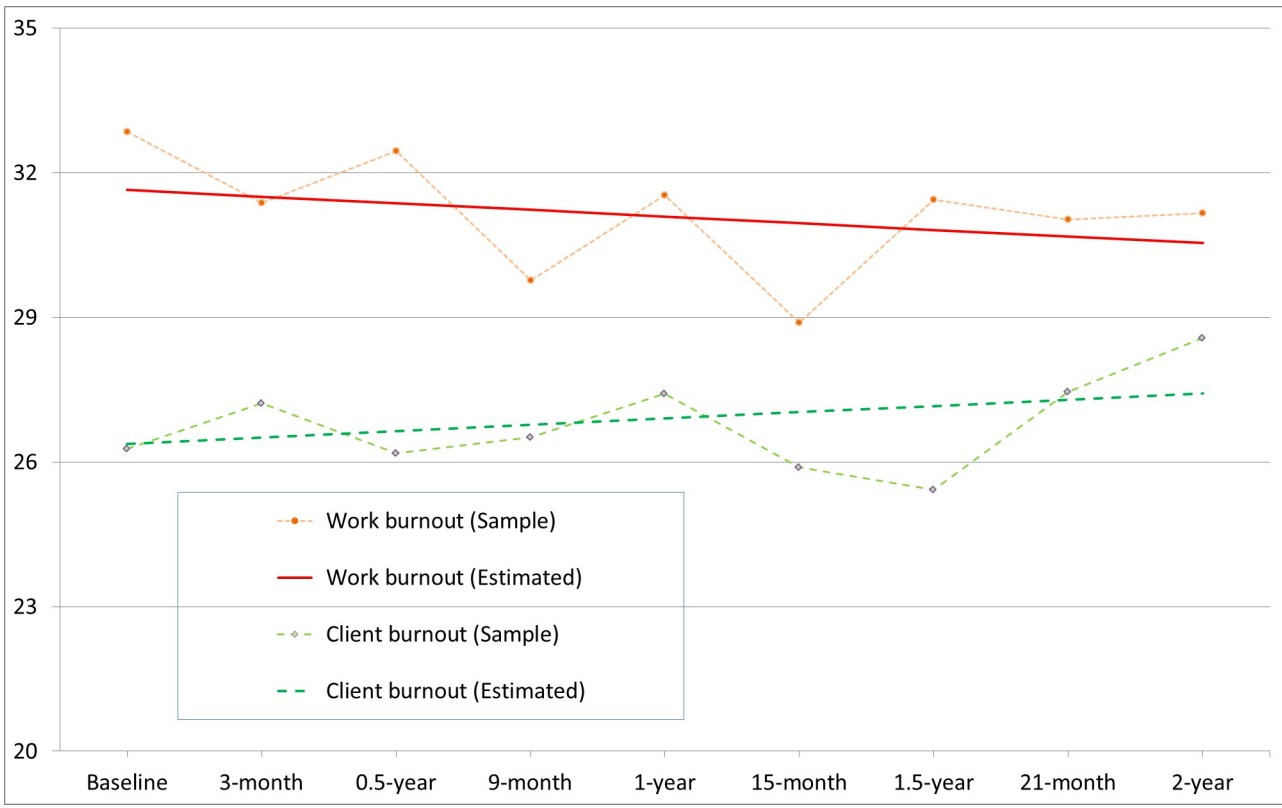

**Fig 2. Developmental trajectories of work burnout and client burnout in the child care workers.**

## Associations between the changes in role stress and burnout

Table 3 shows the regression results of changes in work/client burnout on changes in workers' role stress. The changes in role ambiguity and role conflict significantly and positively predicted the change in work burnout ($\beta$ = 0.31–0.46, $p$ < .01). Client burnout was significantly and positively affected by changes in role conflict ($\beta$ = 0.57, $p$ < .01) but not by changes in role ambiguity ($\beta$ = 0.24, $p$ = .06). The effects of role conflict on work burnout and client burnout were stronger than those of role ambiguity. The model-explained variance was significant for the changes in work burnout and client burnout ($R^2$ = 27.4%–33.3%, $SE$ = 0.08–0.09, $p$ < .01).

## Associations between the changes in role stress and total behavioral problems

The data from the 381 residents were clustered within the data from the 76 care workers, with an average cluster size of 5.01 residents per care worker. For the residents' total behavioral problems, the intraclass correlations between the baseline level and changes in the first 9 months and the subsequent 18 months were 0.187, 0.080, and 0.119, respectively. The changes in role stress significantly and positively predicted changes in behavioral problems ($\beta$ = 0.26–0.29, $p$ < .05) over the first phase. No such effects were found ($p$ = .08–.79) on changes in behavioral problems over the second phase. In the multilevel regression, the model-explained variance was significant for the changes in total behavioral problems over the first phase ($R^2$ = 60.1%, $SE$ = 0.22, $p$ < .01) but not over the second phase ($R^2$ = 46.6%, $SE$ = 0.31, $p$ = .13).

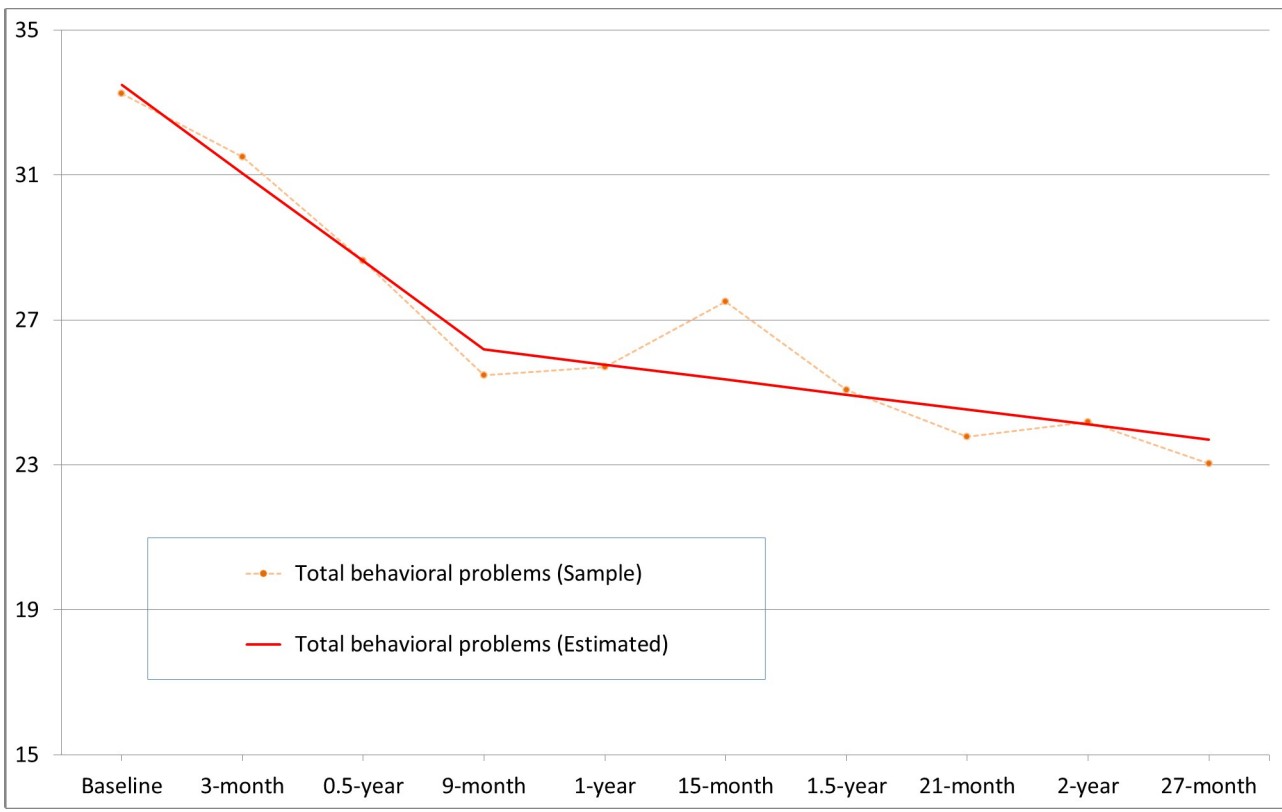

**Fig 3. Developmental trajectories of total behavioral problems in the child residents.**

## Discussion

This longitudinal study with an intensive multi-wave panel design contributes to a better understanding of the developmental trajectories of staff well-being and residents' behavioral problems over a 2-year period. The absence of noticeable trends in the workers' well-being suggests that the overall levels of job demands and burnout did not worsen over the study period. The present sample of child care workers reported higher levels of work burnout and client burnout than were reported in a previous study of mental health care workers in Hong Kong [25]. The temporal stability of burnout might reflect its relatively chronic nature, as suggested by Shirom [33]. The significant variations in the temporal changes in role stress and

**Table 3. Regression estimates of changes in staff burnout and total behavioral problems on changes in role ambiguity/conflict.**

| Outcome variables | Δ Role ambiguity | | | | Δ Role conflict | | | |
|---|---|---|---|---|---|---|---|---|
| | *B* | *SE* | *p* | *β* | *B* | *SE* | *p* | *β* |
| ΔWork burnout | 2.04 | 0.66 | < .01** | 0.31 | 7.48 | 1.83 | < .01** | 0.46 |
| ΔClient burnout | 1.57 | 0.84 | .06 | 0.24 | 9.33 | 1.83 | < .01** | 0.57 |
| ΔTotal behavioral problems | | | | | | | | |
| First 9 months | 1.69 | 0.79 | .03* | 0.29 | 4.97 | 2.32 | .03* | 0.26 |
| 9–27 months | -0.71 | 0.41 | .08 | -0.22 | -0.29 | 1.07 | .79 | -0.05 |

* $p < .05$

** $p < .01$. B: unstandardized regression coefficients; *SE*: standard errors; *β*: standardized regression coefficients.

burnout imply the existence of diverging trajectories among the workers, with some of them reporting worse well-being and others reporting better well-being in the workplace over the study period. This result is similar to the findings of another longitudinal study [34], in which the sample of Chinese mental health care workers displayed significant inter-individual variations in the temporal changes in work climate, burnout, and depression over 2 years.

The significant and positive associations between role stress and burnout suggest that workers who identified increasing ambiguity and conflicts in their roles showed greater increases in work burnout. These results lend empirical support to the longitudinal relationship between role stress and burnout and suggest that the presence of role ambiguity or conflicts could be linked with increasing job burnout among child care workers. Consistent with previous studies [18], role conflict showed a stronger association with burnout than with role ambiguity. This discrepancy appears to suggest a more important role for role conflict in the development of burnout.

The current study presents an original investigation of the linkage between child care workers' role stress and child residents' behavioral problems in the residential care setting. The piecewise developmental trajectories for the residents' behavioral problems imply substantial declines over the first 9 months of residential stay, followed by a leveling-off over the following 18 months. The multilevel regression results found significant linkages between increases in workers' role stress and slower declines in residents' total behavioral problems, which suggests potential deleterious effects of role stress on service quality in terms of clients' well-being. Workers with high job demands may show depersonalized attitudes toward their clients [35]. From a theoretical perspective, role stress could be indirectly associated with residents' well-being via burnout, although the present study could not examine this mediating effect. Future studies should attempt to explore the mediating role of burnout in the relationship between role stress and residents' well-being.

## Practical implications

From a practical point of view, staff orientation programs could offer a realistic overview of the challenges in the child care setting. The socialization process could help organizational newcomers to better understand the organizational values and manage their work expectations. Rai [36] found that organizational values such as procedural justice and organizational commitment reduced role stress. Job ambiguity could be ameliorated via supervisory support and regular team meetings [37, 38]. Leader–member exchange and perceived organizational support have been shown to mediate the relationship between changes in role stress and turnover intention among child welfare workers over a 6-month period [39]. Participatory decision-making and mutual communications between workers and supervisors could be promoted via empowering and authentic leadership [40, 41]. Fostering a work climate [42] that promotes work autonomy, collegial support, and professional visibility could reduce workers' perceived role stress. The creation of decentralized job conditions could promote job autonomy, which has been found to buffer the relationship between role stress and burnout among social workers [43].

In Hong Kong, child residents with emotional problems often need to wait several months or more to receive professional help. The current shortage of counseling services implies delayed intervention and a longer duration of stay in residential homes. The findings highlight the need for timely intervention during the initial phase of residential stay to tackle residents' emotional issues. The provision of counseling services can reduce residents' internalizing and externalizing behavioral problems [44]. Alternative interventions such as play therapy and expressive art therapy [45] facilitate youth residents' emotional expression via non-verbal

means. Their potential effectiveness in improving residents' well-being could be examined in future studies. The introduction of counselor posts in the residential setting could provide continuous counseling support for those in need and would benefit residents' development and social integration. This could shorten the average length of stay and free up valuable residential spots for those with placement needs. For existing child workers, a clear division of labor would enable them to focus on their day-to-day caring duties, which could reduce their role stress and alleviate the issue of burnout.

## Study limitations

The present study has several limitations. First, the study was based on a small sample of child care workers in residential care homes. Given the potential concerns about the representativeness of the sample, caution is warranted in generalizing the findings to other fields. Second, the small sample of workers did not allow an evaluation of the assumption of the measurement instruments' invariance across time. The potential non-invariance biases in item loadings and intercepts may cast doubt on the comparisons of means across time. Future studies should attempt to recruit a larger sample of workers to examine the longitudinal invariance of the scales.

Third, the measures of staff well-being and residents' behavioral problems were both completed by the workers. Common method variance may have biased the parameter estimates. Future research could include alternative measures such as work performance appraised by supervisors. Fourth, the study included role stress as an example of job demands but did not include a measure of job resources. Social support from supervisors and coworkers has been shown to buffer the impact of job demands on burnout [46, 47]. Further studies should assess the developmental trajectories of work engagement [48] as a positive workplace outcome.

## Supporting information

**S1 File. SPSS dataset analyzed in the present study.**
(SAV)

## Acknowledgments

We express our gratitude to the staff and child residents of the four non-governmental organizations (The Boys' and Girls' Club Association of Hong Kong, Evangel Children's Home, Home Care for Girls, and Hong Kong Christian Service) for their participation in the study. We thank Irene Cheung, Kathy Chau, Winnie Lam, and Teresa Chiu for their help in the implementation and coordination of this study.

## Author Contributions

**Conceptualization:** Ted C. T. Fong, Rainbow T. H. Ho.

**Data curation:** Ted C. T. Fong.

**Formal analysis:** Ted C. T. Fong.

**Funding acquisition:** Rainbow T. H. Ho.

**Investigation:** Ted C. T. Fong, Joyce C. Y. Fong.

**Methodology:** Ted C. T. Fong, Rainbow T. H. Ho.

**Project administration:** Joyce C. Y. Fong.

**Resources:** Rainbow T. H. Ho.

**Software:** Ted C. T. Fong.

**Supervision:** Rainbow T. H. Ho.

**Visualization:** Ted C. T. Fong.

**Writing – original draft:** Ted C. T. Fong.

**Writing – review & editing:** Rainbow T. H. Ho, Joyce C. Y. Fong.

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
