## [Decision Letter · Decision Letter 0]

20 Apr 2021

PONE-D-21-04372

Effects of role ambiguity and role conflict on staff burnout and residents’ behavioral problems: A 2-year longitudinal study in child care homes in Hong Kong

PLOS ONE

Dear Dr. Fong,

Thank you for submitting your manuscript to PLOS ONE. After careful consideration, we feel that it has merit but does not fully meet PLOS ONE’s publication criteria as it currently stands. Therefore, we invite you to submit a revised version of the manuscript that addresses the points raised during the review process.

We look forward to receiving your revised manuscript.

Kind regards,

Gabriela Topa, Ph. D.

Academic Editor

PLOS ONE

Journal Requirements:

This is  cross sectional observational study, as such, we do not feel that any conclusions on the effects of role ambiguity and conflict   can be supported; thus, we ask that you revise the text (especially, but no limited to, the title) to avoid unsupported statements.

Please provide additional information regarding the considerations  made for the children in care included in this study during the informed consent procedure.

Additional Editor Comments (if provided):

Reviewers' comments:

Reviewer's Responses to Questions

**Comments to the Author**

1. Is the manuscript technically sound, and do the data support the conclusions?

Reviewer #1: Yes

Reviewer #2: Yes

2. Has the statistical analysis been performed appropriately and rigorously? 

Reviewer #1: I Don't Know

Reviewer #2: Yes

3. Have the authors made all data underlying the findings in their manuscript fully available?

Reviewer #1: Yes

Reviewer #2: Yes

4. Is the manuscript presented in an intelligible fashion and written in standard English?

Reviewer #1: Yes

Reviewer #2: Yes

5. Review Comments to the Author

Reviewer #1: This is an interesting topic; however, I have some critical concerns and provide some comments that might help the authors to develop their paper.

About the theoretical introduction:

The manuscript contains several statements that need to be supplemented with references, e.g.:

line 49: …..and could result in depersonalized working attitudes towards the clients.

In fact, there is some research missing related to its foundation and model and that should be mentioned in the theory section. In my opinion, one of the main concerns is that the authors do not sufficiently elaborate their main ideas and hypotheses. A clear theoretical perspective is lacking, especially regarding the assumptions of the effects of role ambiguity and role conflict on staff burnout on resident behavior problems.

Method:

In your current participants and procedures section it would be nice to have more description about if incentives were offered to the participants. Finally, it would nice to report the total response rate across the time points. Adding a table with descriptive demographic statistics would make it easier for the reader.

Discussion:

You must explain the reasons for some results of your research, for example:

“The significant inter-individual variations in the temporal changes denote the existence 256 of diverging trajectories of role stress and burnout among the workers. Some workers 257 reported increasing levels of role stress while others reported declining levels of role stress 258 over the study period”.You should discuss the results of your research and compare them with previous research, providing references and explanations for the differences between the results. The concern is that it most of the section discusses how the current study confirms already established relationships which leaves the reader wondering what is the unique contribution of this study.

It should be noted that the mediating role of burnout in the relationship between role stress and the well-being of residents was not possible in the present study.

I am not a native English speaker; however my impression is that the manuscript would benefit from a substantial editing in terms of grammar, syntax, and wording by a native English speaker.

Reviewer #2: Dear authors,

Thank you so much for submitting your interesting work. It is a very well-conducted investigation within one emotionally vulnerable work sector. The children and adolescents' behaviour problems are related to undesirable processes like role ambiguity, role conflict, and burnout.

However, I would like to ask some questions which appeared in my mind, and I hope you find them interesting:

- The authors describe the scales used in the investigation, supplying necessary information such as levels of reliability or the scoring system. Concerning this, I wonder if some of these questionnaires define levels based on the scores (e.g. high role ambiguity over X points)

- L271-L277. The authors mention some interventions for residents, but I am not sure if those are present in the residential care homes in Hong Kong or are just a reference about alternative ways to enhance the participants' behaviour. More information about residents (e.g. which kind of cares offered to them) could be useful and may be presented in the introduction. It would make the introduction and conclusions (like L271-L274 more linked).

Thank you so much,

Best wishes

6. PLOS authors have the option to publish the peer review history of their article (what does this mean?). If published, this will include your full peer review and any attached files.

Reviewer #1: No

Reviewer #2: No

---

## [Author Response · Author response to Decision Letter 0]

31 May 2021

Response to Reviewers [PONE-D-21-04372.R1]

Thank you very much for the valuable comments on the manuscript. We have studied all of the comments carefully and revised the manuscript accordingly. Please refer to the point-by-point response (underlined and starting with “>>”) below to each point. The major changes in the manuscript have been highlighted in red, blue, and green for the journal requirement, reviewer 1, and reviewer 2, respectively, for easier reference.

Journal Requirements:

1. Please ensure that your manuscript meets PLOS ONE's style requirements, including those for file naming. The PLOS ONE style templates can be found at journals.plos.org/plosone/s/file?id=wjVg/PLOSOne_formatting_sample_main_body.pdf journals.plos.org/plosone/s/file?id=ba62/PLOSOne_formatting_sample_title_authors_affiliations.pdf

>> We have formatted the manuscript to the PLOS ONE’s style requirement.

2. This is cross sectional observational study, as such, we do not feel that any conclusions on the effects of role ambiguity and conflict can be supported; thus, we ask that you revise the text (especially, but not limited to, the title) to avoid unsupported statements. 

>> Thank you for the remarks. Please note that the present study actually has a 2-year longitudinal design. However, since all of the study variables were assessed simultaneously in this study, we agree that there could be reciprocal effects from staff burnout and residents’ behavioral problems to role stress. We have revised the text (especially the title and abstract) to avoid focusing on the “effects” of role stress.

3. Please provide additional information regarding the considerations made for the children in care included in this study during the informed consent procedure.

>> We have provided more detailed information regarding the ethical considerations for the child residents in terms of informed consent under the “Ethical considerations” in page 7.

>> We have reviewed the reference list to ensure that it is complete and correct. No retracted articles have been cited in this manuscript.

5. Review Comments to the Author

Reviewer #1: This is an interesting topic; however, I have some critical concerns and provide some comments that might help the authors to develop their paper.

About the theoretical introduction: The manuscript contains several statements that need to be supplemented with references, e.g.: line 49: …..and could result in depersonalized working attitudes towards the clients. In fact, there is some research missing related to its foundation and model and that should be mentioned in the theory section. 

>> We have added several more references in the Introduction and in line 49 about the statement on depersonalized working attitudes to give a broader background of the context in child care services. The added references include (Whittaker, 2016), (Kim & Lee, 2009), (Demerouti et al., 2001), (Kim, 2011), (Boyas, Wind & Ruiz, 2015).

In my opinion, one of the main concerns is that the authors do not sufficiently elaborate their main ideas and hypotheses. A clear theoretical perspective is lacking, especially regarding the assumptions of the effects of role ambiguity and role conflict on staff burnout on resident behavioral problems.

>> We have elaborated more on the main ideas regarding the assumptions of the effects of role stress (role ambiguity and role conflict) on staff burnout in the third paragraph in page 4 and for residents’ behavioural problems in the fifth paragraph in page 5. We have reorganized the Introduction section with the following flow: 

1st paragraph: Context of child care services in Hong Kong; 

2nd paragraph: Job demands and burnout in child care workers;

3rd paragraph: The Job Demands-Resources model and role ambiguity and role conflict 

4th paragraph: Associations between role stress and burnout and research gap

5th paragraph: Potential association between role stress and residents’ behavioural problems 

6th paragraph: Research objectives and hypotheses 

Method: In your current participants and procedures section it would be nice to have more description about if incentives were offered to the participants. Finally, it would be nice to report the total response rate across the time points. Adding a table with descriptive demographic statistics would make it easier for the reader.

>> We have clarified in Ethical considerations section in page 7 that no incentives were offered to the participants. A new table (Table 1) has been added to report the retention rates of the participants across the time points in the “Attrition analysis” section under Results. 

Discussion: You must explain the reasons for some results of your research, for example:

“The significant inter-individual variations in the temporal changes denote the existence 256 of diverging trajectories of role stress and burnout among the workers. Some workers 257 reported increasing levels of role stress while others reported declining levels of role stress 258 over the study period”. You should discuss the results of your research and compare them with previous research, providing references and explanations for the differences between the results. 

>> Thank you for the comment. We have added a reference of (Fong et al, 2016) to compare our results of inter-individual variations with the previous findings in the end of the first paragraph in the Discussion section.

The concern is that it most of the section discusses how the current study confirms already established relationships which leaves the reader wondering what is the unique contribution of this study.

>> Apart from discussions on how the current findings compare to established relationships, we have delineated the contribution of this study. The main contributions are 1) an intensive multi-wave panel design which leads to better understanding of the developmental trajectories of role stress and burnout in child care workers and behavioral problems of child residents over the 2-year period, 2) empirical support to longitudinal relationships between role stress and burnout, 3) original investigation of the linkage between workers’ role stress and child residents’ behavioural problems.

It should be noted that the mediating role of burnout in the relationship between role stress and the well-being of residents was not possible in the present study.

>> Thank you for the comment. We have noted that the mediating role of burnout in the relationship between role stress and residents’ well-being could not be examined in the present study in the 3rd paragraph of the Discussion section.

I am not a native English speaker; however my impression is that the manuscript would benefit from a substantial editing in terms of grammar, syntax, and wording by a native English speaker.

>> Thank you for the advice. The revised manuscript has been sent to academic editing by a native English speaker for editing of grammar and syntax.

Reviewer #2: Thank you so much for submitting your interesting work. It is a very well-conducted investigation within one emotionally vulnerable work sector. The children and adolescents' behaviour problems are related to undesirable processes like role ambiguity, role conflict, and burnout. However, I would like to ask some questions which appeared in my mind, and I hope you find them interesting:

- The authors describe the scales used in the investigation, supplying necessary information such as levels of reliability or the scoring system. Concerning this, I wonder if some of these questionnaires define levels based on the scores (e.g. high role ambiguity over X points)

>> Thank you for the advice. There are no known cutoff scores for role ambiguity and role conflict of the workers and total behavioral problems of the child residents. We have added the cutoff criteria for high degree of burnout for the CBI under the Measure section in page 9, the Results section in page 13, and the beginning of the Discussion section in page 15.

- L271-L277. The authors mention some interventions for residents, but I am not sure if those are present in the residential care homes in Hong Kong or are just a reference about alternative ways to enhance the participants' behaviour. More information about residents (e.g. which kind of cares offered to them) could be useful and may be presented in the introduction. It would make the introduction and conclusions (like L271-L274 more linked).

>> Those interventions are currently not present in residential care homes in Hong Kong. They are provided as a reference about and alternative ways of interventions. We have moved these sentences to the 2nd paragraph “Practical implications” section in page 18, which has been revised to describe the context of residential care offered to the child residents in Hong Kong.

---

## [Decision Letter · Decision Letter 1]

13 Jun 2022

Temporal relationships among role stress, staff burnout, and residents’ behavioral problems: A 2-year longitudinal study in child care homes in Hong Kong

PONE-D-21-04372R1

Dear Dr. Ho,

We’re pleased to inform you that your manuscript has been judged scientifically suitable for publication and will be formally accepted for publication once it meets all outstanding technical requirements.

Kind regards,

Muhammad A. Z. Mughal, PhD

Academic Editor

PLOS ONE

Additional Editor Comments (optional):

Reviewers' comments:

Reviewer's Responses to Questions

**Comments to the Author**

1. If the authors have adequately addressed your comments raised in a previous round of review and you feel that this manuscript is now acceptable for publication, you may indicate that here to bypass the “Comments to the Author” section, enter your conflict of interest statement in the “Confidential to Editor” section, and submit your "Accept" recommendation.

Reviewer #1: All comments have been addressed

2. Is the manuscript technically sound, and do the data support the conclusions?

Reviewer #1: Yes

3. Has the statistical analysis been performed appropriately and rigorously? 

Reviewer #1: Yes

4. Have the authors made all data underlying the findings in their manuscript fully available?

Reviewer #1: Yes

5. Is the manuscript presented in an intelligible fashion and written in standard English?

Reviewer #1: Yes

6. Review Comments to the Author

Reviewer #1: I have reviewed the suggested changes to the authors of the paper. The changes have been resolved successfully.

7. PLOS authors have the option to publish the peer review history of their article (what does this mean?). If published, this will include your full peer review and any attached files.

Reviewer #1: No

---

## [Editor Report · Acceptance letter]

13 Jul 2022

PONE-D-21-04372R1 

Temporal relationships among role stress, staff burnout, and residents’ behavioral problems: A 2-year longitudinal study in child care homes in Hong Kong 

Dear Dr. Ho:

I'm pleased to inform you that your manuscript has been deemed suitable for publication in PLOS ONE. Congratulations! Your manuscript is now with our production department. 

Kind regards, 

on behalf of

Dr. Muhammad A. Z. Mughal 

Academic Editor

PLOS ONE